optics/light microscopy/crystallography

chirality, optical activity, polarization, polarimetry, microscopy, DIC

**Author for correspondence:**
R. P. Cameron
e-mail: robert.p.cameron@strath.ac.uk

# Interference-contrast optical activity: a new technique for probing the chirality of anisotropic samples and more

## R. P. Cameron[1], U. Vogl[2] and N. Trautmann[2]

[1]SUPA and Department of Physics, University of Strathclyde, Glasgow, G4 0NG, UK
[2]Corporate Research and Technology, Carl Zeiss AG, Carl-Zeiss-Strasse 22, Oberkochen 73447, Germany

RPC, 0000-0002-8809-5459; UV, 0000-0003-2399-2797

We introduce interference-contrast optical activity (ICOA) as a new technique for probing the chirality of anisotropic samples and more. ICOA could underpin a new class of 'chiral microscopes', with potential applications spanning the range of chirality and beyond. Two possible versions of ICOA are described explicitly; one designed to probe the optical rotation of a transparent sample *regardless* of the sample's linear birefringence (ICOA-OR) and another designed to probe *gradients* in the optical rotation of a transparent sample (ICOA-GOR). Simulated results for $\alpha$-quartz lead us to suggest that ICOA-GOR might be applied to help monitor the growth of chiral crystals in the pharmaceutical industry. Possible directions for future research are highlighted.

## 1. Introduction

Optical rotation, circular dichroism and other manifestations of optical activity are measured routinely for isotropic samples, serving as hallmarks of chirality in applications ranging from the determination of sugar concentrations to the investigation of virus structures [1–4]. By contrast, measurements of optical activity are seldom reported for anisotropic samples, the main reason being that anisotropic samples usually exhibit linear birefringence, linear dichroism and other effects that convolve with and partially suppress optical activity [5–9]. According to one source: 'Measuring [optical rotation] and [circular dichroism] in crystals of arbitrary symmetry has for two centuries been likened to the search for a needle in a haystack. For this reason, we know virtually nothing

from experiment about the orientational dependence of chiroptics of molecules, an enormous hole in the science of molecular chirality.' [9].

In this paper, we introduce interference-contrast optical activity (ICOA) as a new technique for probing the chirality of anisotropic samples and more. ICOA could underpin a new class of 'chiral microscopes', with potential applications spanning the range of chirality and beyond; chirality is sufficient but not always necessary for the presence of optical activity, which is exhibited naturally by certain achiral anisotropic samples and can also be induced in a sample by certain influences such as a static magnetic field [2,6,8,10–12]. In what follows, we describe two possible versions of ICOA explicitly; one designed to probe the optical rotation of a transparent sample *regardless* of the sample's linear birefringence (ICOA-OR) and another designed to probe *gradients* in the optical rotation of a transparent sample (ICOA-GOR). ICOA-OR and ICOA-GOR are distinct from all polarimetric techniques known to the authors at the time of writing, including HAUP-based techniques [7,10,11,13,14], existing polarization interferometry techniques [15], optical heterodyne polarimetry [16,17], Metripol-based techniques [7,18], CRDP and other such cavity-based techniques [19–21] and Mueller matrix polarimetry [9,22]. ICOA-GOR, in particular, has elements in common with, but is subtly distinct from, DIC-based techniques [23–29]. ICOA is complementary to chiral rotational spectroscopy; a technique proposed recently by one of the authors for determining orientated chiroptical information about individual molecules [30].

In what follows, we consider ourselves to be in an inertial frame of reference described by right-handed Cartesian coordinates $x$, $y$ and $z$ with associated unit vectors $\hat{\mathbf{x}}$, $\hat{\mathbf{y}}$ and $\hat{\mathbf{z}}$. Rotations are dictated by the left-hand rule, with optical rotations taken about the direction of propagation of the light. We work in the domain of classical optics using the Jones vector formalism [2,22,31,32], with the upper components of our Jones vectors corresponding to the $x$ component of the electric field and the lower components corresponding to the $y$ component.

## 2. ICOA-OR

In this section, we describe ICOA-OR; a version of ICOA designed to probe the optical rotation of a transparent sample *regardless* of the sample's linear birefringence. For the sake of concreteness, we consider the basic set-up depicted schematically in figure 1, modelled as described below. Other set-ups capable of achieving the same results are conceivable.

A light source $\mathcal{L}$ produces an initial (optical) field in the form of weak, planar, monochromatic light of angular frequency $\omega$ and diagonal polarization ($D$). The initial field first propagates through a variable wave plate $\mathcal{VW}$ with axes aligned vertically and horizontally, rendering it elliptically polarized ($E$) in general. A polarizing beam splitter $\mathcal{BS}_1$ together with a polarization-independent mirror $\mathcal{M}_1$ then divides the initial field into a vertically polarized ($V$) sampling field and a horizontally polarized ($H$) reference field, with transverse separation described by the shear vector $\mathbf{s} = s_x\hat{\mathbf{x}} + s_y\hat{\mathbf{y}}$. The sampling and reference fields first propagate through polarization rotators $\mathcal{PR}_1$ and $\mathcal{PR}_2$ set to rotate through an angle $\sigma$, resulting in the Jones vectors

$$\tilde{J}_{\mathrm{s}}^{(\sigma)} = \frac{1}{\sqrt{2}}\left[\cos\sigma,\ -\sin\sigma\right]^{\mathrm{T}} \tag{2.1}$$

and

$$\tilde{J}_{\mathrm{r}}^{(\sigma)} = \frac{1}{\sqrt{2}}\left[\sin\sigma,\ \cos\sigma\right]^{\mathrm{T}}. \tag{2.2}$$

They then propagate through a sample zone, where a thin, transparent sample $\mathcal{S}$ is located in the path of the sampling field. The sample is refractive-index matched with the surrounding medium, taken to be an achiral, transparent fluid with refractive index $n$. After the sample zone the Jones vector of the sampling field is $\tilde{M}\tilde{J}_{\mathrm{s}}^{(\sigma)}$, where $\tilde{M}$ is a Jones matrix embodying the optical properties of the sample;

$$\tilde{M} = \mathrm{e}^{\mathrm{i}\alpha\Delta z}\begin{bmatrix} \cos(\tau\Delta z) + \mathrm{i}\delta\Delta z\,\mathrm{sinc}(\tau\Delta z) & (\mathrm{i}\beta + \gamma)\Delta z\,\mathrm{sinc}(\tau\Delta z) \\ (\mathrm{i}\beta - \gamma)\Delta z\,\mathrm{sinc}(\tau\Delta z) & \cos(\tau\Delta z) - \mathrm{i}\delta\Delta z\,\mathrm{sinc}(\tau\Delta z) \end{bmatrix} \tag{2.3}$$

with

$$\tau = \sqrt{\beta^2 + \gamma^2 + \delta^2}, \tag{2.4}$$

where $\alpha = \alpha(x,y)$ accounts for the mean refractive index of the sample (relative to $n$), $\beta = \beta(x,y)$ accounts for the diagonal-antidiagonal linear birefringence of the sample, $\gamma = \gamma(x,y)$ accounts for the circular birefringence of

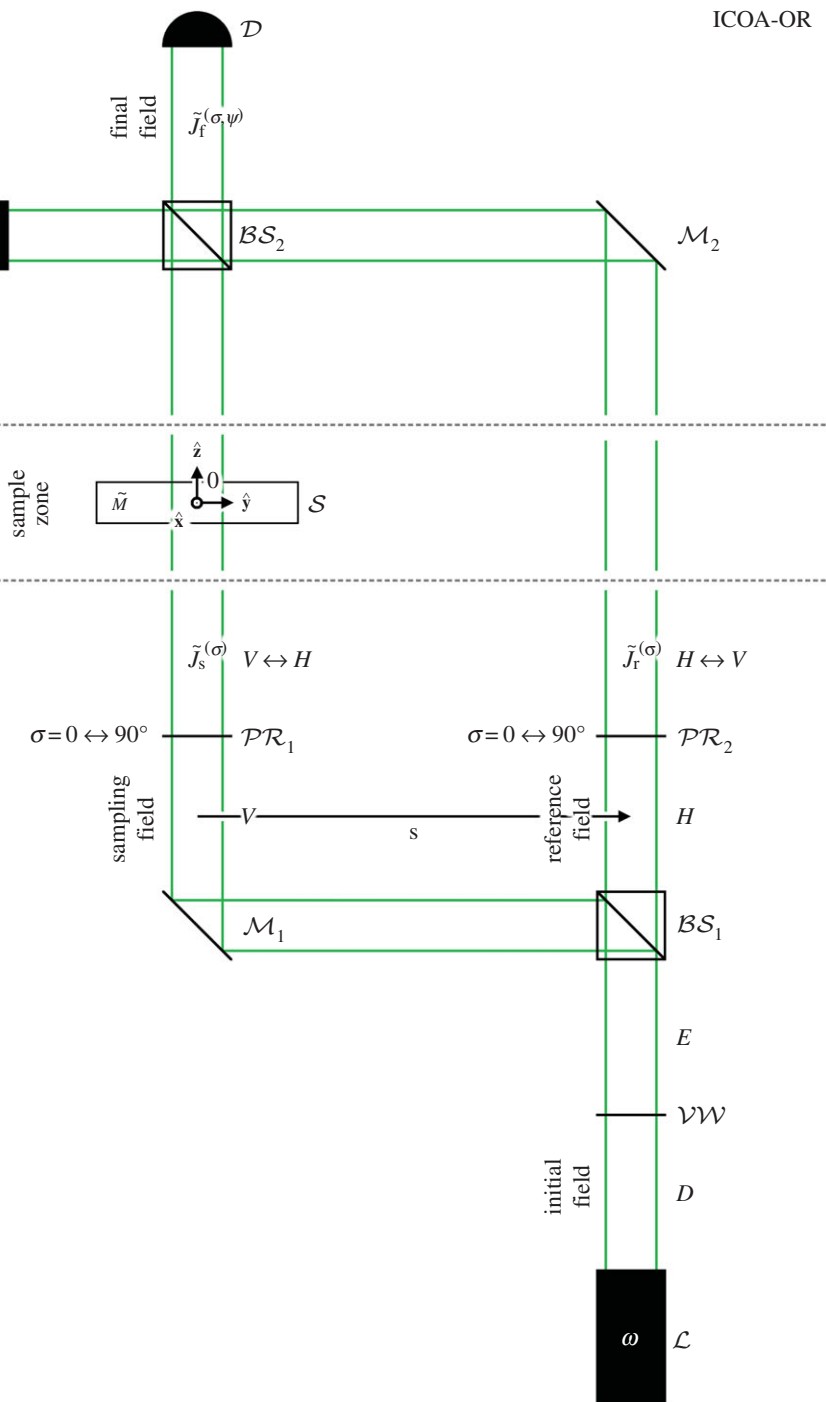

**Figure 1.** A basic set-up for ICOA-OR, depicted schematically (with a shear vector of the form $\mathbf{s} = |\mathbf{s}|\hat{\mathbf{y}}$).

the sample, $\delta = \delta(x, y)$ accounts for the vertical–horizontal linear birefringence of the sample and $\Delta z = \Delta z(x, y)$ is the geometrical thickness of the sample [2,22,31,32]. The sampling and reference fields are recombined by a polarization-independent mirror $\mathcal{M}_2$ together with a polarization-independent 50 : 50 beam splitter $\mathcal{BS}_2$, producing a final field with Jones vector

$$\tilde{J}_{\mathrm{f}}^{(\sigma,\psi)} = \frac{1}{\sqrt{2}}\left[\tilde{M}\tilde{J}_{\mathrm{s}}^{(\sigma)} + \mathrm{e}^{\mathrm{i}\psi}\tilde{J}_{\mathrm{r}}^{(\sigma)}\right], \tag{2.5}$$

where $\psi$ is a relative phase, tunable via $\mathcal{VW}$. A detector $\mathcal{D}$ records the intensity

$$I_{\mathrm{OR}}^{(\sigma,\psi)} = \tilde{J}_{\mathrm{f}}^{(\sigma,\psi)\dagger}\tilde{J}_{\mathrm{f}}^{(\sigma,\psi)}, \tag{2.6}$$

of the final field, normalized here to give values in the range [0, 1]. Let us suppose now that $\mathcal{PR}_1$ and $\mathcal{PR}_2$ are modulated between their $\sigma = 0$ and $\sigma = 90°$ settings such that the (average) intensity of the final field becomes

$$\bar{I}_{OR}^{(\psi)} = \frac{1}{2} \left[ I_{OR}^{(0,\psi)} + I_{OR}^{(90°,\psi)} \right]$$
$$= \frac{1}{2} \left[ 1 + \bar{C}_{OR}^{(\psi)} \right] \tag{2.7}$$

with

$$\bar{C}_{OR}^{(\psi)} = -\cos(\alpha\Delta z - \psi)\Delta\theta \, \text{sinc}(\tau\Delta z), \tag{2.8}$$

where $\Delta\theta = \gamma\Delta z$ is the (bare) optical rotation of $\mathcal{S}$. Equation (2.8) is the (average) contrast due to interference of the sampling and reference fields in the final field. According to equation (2.8), the contrast can be non-zero ($\bar{C}_{OR}^{(\psi)} \neq 0$) if and only if the optical rotation is non-zero ($\Delta\theta \neq 0$), *regardless* of the linear birefringence ($\beta$ and $\delta$) of $\mathcal{S}$.

If the sample $\mathcal{S}$ is sufficiently thin and well matched with the surrounding medium that $|\alpha|\Delta z \lesssim 1$ and $\tau\Delta z \lesssim 1$, equation (2.8) reduces to

$$\bar{C}_{OR}^{(\pi)} \approx \Delta\theta, \tag{2.9}$$

for a choice of relative phase equivalent to $\psi = \pi$. According to equations (2.7) and (2.9), the intensity $\bar{I}_{OR}^{(\pi)}$ embodies an image of the optical rotation $\Delta\theta$, with regions of positive optical rotation ($\Delta\theta > 0$) appearing brightened ($\bar{C}_{OR}^{(\pi)} > 0$) and regions of negative optical rotation ($\Delta\theta < 0$) appearing darkened ($\bar{C}_{OR}^{(\pi)} < 0$), *regardless* of the linear birefringence ($\beta$ and $\delta$) of $\mathcal{S}$. If $\mathcal{S}$ is not sufficiently thin and well matched with the surrounding medium to satisfy $|\alpha|\Delta z \lesssim 1$ but is nevertheless sufficiently flat that $|\Delta(\alpha\Delta z)| \lesssim 1$, the desired image can be obtained for a choice of relative phase equivalent to $\psi = (\alpha\Delta z + \pi)\text{mod}(2\pi)$ (assuming that $\tau\Delta z \lesssim 1$). If neither $|\alpha|\Delta z \lesssim 1$ nor $|\Delta(\alpha\Delta z)| \lesssim 1$ is satisfied, appropriate compensations might be made by deforming the phase fronts of the sampling and/or reference fields using adaptive optics [33,34].

The physical origin of equation (2.9) can be understood simply as follows, where we assume that $|\alpha|\Delta z \ll 1$ and $\tau\Delta z \ll 1$. For a given setting of the polarization rotators $\mathcal{PR}_1$ and $\mathcal{PR}_2$ ($\sigma = 0$ or $\sigma = 90°$), non-zero optical rotation ($\Delta\theta \neq 0$) yields an electric-field component in the sampling field aligned with the electric field of the reference field and the two interfere in the final field, either constructively ($\bar{C}_{OR}^{(\pi)} > 0$) or destructively ($\bar{C}_{OR}^{(\pi)} < 0$) depending on the sign of the optical rotation ($\Delta\theta > 0$ or $\Delta\theta < 0$). Non-zero diagonal-antidiagonal linear birefringence ($\beta \neq 0$) also yields an electric-field component in the sampling field aligned with the electric field of the reference field; however, the two do not (strongly) interfere in the final field as they differ in phase by (approximately) a quarter cycle. Modulating $\mathcal{PR}_1$ and $\mathcal{PR}_2$ between their $\sigma = 0$ and $\sigma = 90°$ settings provides an extra layer of immunity to non-zero linear birefringence (in particular, $\beta \neq 0$) as it is physically equivalent to rotating the sample $\mathcal{S}$ back and forth by a quarter turn about the direction of propagation of the light, effectively modulating the signs of the linear birefringence parameters $\beta$ and $\delta$. Certain undesirable terms vanish accordingly, in turn relaxing requirements on the mean refractive index parameter $\alpha$ and the geometrical thickness $\Delta z$.

# 3. ICOA-GOR

In this section, we describe ICOA-GOR; a version of ICOA designed to probe *gradients* in the optical rotation of a transparent sample. For the sake of concreteness, we consider the basic set-up depicted schematically in figure 2, modelled as described below. This set-up was inspired by the lens-free microscope described in [29]. Other set-ups capable of achieving the same results are conceivable.

The set-up is similar to that considered in §2 for ICOA-OR but differs in the following key respects. The initial field is divided into sampling and reference fields with a small angular separation $\zeta$ using a Sénarmont prism $\mathcal{SP}$. The sampling and reference fields propagate through a single polarization rotator $\mathcal{PR}$ set to rotate through an angle $\sigma$, resulting in the Jones vectors

$$\tilde{J}_s^{(\sigma)} = \frac{1}{\sqrt{2}} \left[ \cos\sigma, \ -\sin\sigma \right]^T \tag{3.1}$$

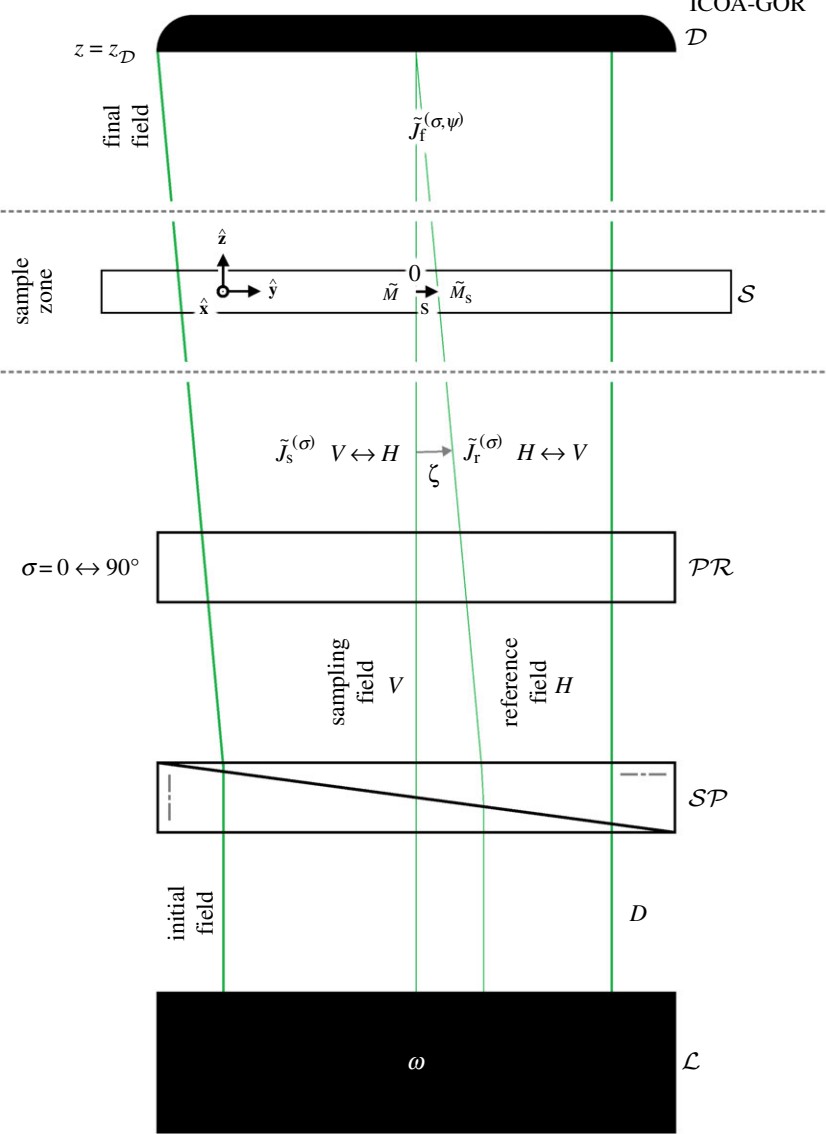

**Figure 2.** A basic set-up for ICOA-GOR, depicted schematically (with a shear vector of the form $\mathbf{s} = |\mathbf{s}|\hat{\mathbf{y}}$).

and

$$\tilde{J}_r^{(\sigma)} = e^{-i\omega n \zeta y/c} \frac{1}{\sqrt{2}} \left[ \sin\sigma, \ \cos\sigma \right]^{\text{T}}. \tag{3.2}$$

The sampling and reference fields both propagate through the sample $\mathcal{S}$, producing a final field with Jones vector

$$\tilde{J}_f^{(\sigma,\psi)}(z = z_{\mathcal{D}}) = \tilde{M}\tilde{J}_s^{(\sigma)} + e^{i\psi}\tilde{M}_s\tilde{J}_r^{(\sigma)}, \tag{3.3}$$

at the position $z = z_{\mathcal{D}}$ of the detector $\mathcal{D}$, where the relative phase $\psi$ is tunable by translating $\mathcal{SP}$ and a subscript $\mathbf{s}$ indicates that a quantity is evaluated at $x + s_x$ and $y + s_y$ rather than $x$ and $y$, thus pertaining to the reference field rather than the sampling field. The intensity of the final field follows as:

$$I_{\text{GOR}}^{(\sigma,\psi)}(z = z_{\mathcal{D}}) = \frac{1}{2}\tilde{J}_f^{(\sigma,\psi)\dagger}(z = z_{\mathcal{D}})\tilde{J}_f^{(\sigma,\psi)}(z = z_{\mathcal{D}}), \tag{3.4}$$

normalized here to give values in the range [0, 1]. Let us suppose now that $\mathcal{PR}$ is modulated between its $\sigma = 0$ and $\sigma = 90°$ settings such that the intensity of the final field becomes

$$\begin{aligned}
\bar{I}_{\text{GOR}}^{(\psi)} &= \frac{1}{2}\left[ I_{\text{GOR}}^{(0,\psi)}(z = z_{\mathcal{D}}) + I_{\text{GOR}}^{(90°,\psi)}(z = z_{\mathcal{D}}) \right] \\
&= \frac{1}{2}\left[ 1 + \bar{C}_{\text{GOR}}^{(\psi)} \right]
\end{aligned} \tag{3.5}$$

with

$$\bar{C}_{\mathrm{GOR}}^{(\psi)} = \cos\left(\alpha_{\mathrm{s}}\Delta z_{\mathrm{s}} - \alpha\Delta z + \psi\right)\Big[\Delta\theta_{\mathrm{s}}\cos\left(\tau\Delta z\right)\mathrm{sinc}(\tau_{\mathrm{s}}\Delta z_{\mathrm{s}}) - \Delta\theta\cos\left(\tau_{\mathrm{s}}\Delta z_{\mathrm{s}}\right)\mathrm{sinc}(\tau\Delta z)$$

$$+ (\beta_{\mathrm{s}}\delta - \beta\delta_{\mathrm{s}})\Delta z_{\mathrm{s}}\Delta z\,\mathrm{sinc}(\tau_{\mathrm{s}}\Delta z_{\mathrm{s}})\,\mathrm{sinc}(\tau\Delta z)\Big], \tag{3.6}$$

where we have assumed that attention is restricted to a small area and taken $\mathrm{e}^{-\mathrm{i}\omega n\zeta y/c} \to 1$ accordingly. Equations (3.5) and (3.6) reduce to equations (2.7) and (2.8) for $\Delta z_{\mathrm{s}} = \alpha_{\mathrm{s}} = \beta_{\mathrm{s}} = \gamma_{\mathrm{s}} = \delta_{\mathrm{s}} = 0$, as they should.

If the sample $\mathcal{S}$ is sufficiently thin and well matched with the surrounding medium that $|\alpha_{\mathrm{s}}\Delta z_{\mathrm{s}} - \alpha\Delta z| \lesssim 1$, $\tau_{\mathrm{s}}\Delta z_{\mathrm{s}} \lesssim 1$ and $\tau\Delta z \lesssim 1$, equation (3.6) reduces to

$$\bar{C}_{\mathrm{GOR}}^{(0)} \approx \Delta\theta_{\mathrm{s}} - \Delta\theta + (\beta_{\mathrm{s}}\delta - \beta\delta_{\mathrm{s}})\Delta z_{\mathrm{s}}\Delta z, \tag{3.7}$$

for a choice of relative phase equivalent to $\psi = 0$. The final term on the right-hand side of equation (3.7) vanishes for many samples of interest (including all of the samples considered in §4), in which case equation (3.7) reduces further still to

$$\bar{C}_{\mathrm{GOR}}^{(0)} \approx \Delta\theta_{\mathrm{s}} - \Delta\theta$$
$$\approx \mathbf{s} \cdot \boldsymbol{\nabla}(\Delta\theta), \tag{3.8}$$

where we have assumed that the shear distance $|\mathbf{s}|$ is small relative to the length scale over which variations in the optical rotation $\Delta\theta$ occur. According to equations (3.5) and (3.8), the intensity $\bar{I}_{\mathrm{GOR}}^{(0)}$ embodies an image of the *gradient* $\mathbf{s} \cdot \boldsymbol{\nabla}(\Delta\theta)$ of $\Delta\theta$ along the shear vector $\mathbf{s}$, with regions of positive gradient $(\mathbf{s} \cdot \boldsymbol{\nabla}(\Delta\theta) > 0)$ appearing brightened $(\bar{C}_{\mathrm{GOR}}^{(0)} > 0)$ and regions of negative gradient $(\mathbf{s} \cdot \boldsymbol{\nabla}(\Delta\theta) < 0)$ appearing darkened $(\bar{C}_{\mathrm{GOR}}^{(0)} < 0)$. Under the same conditions, equation (3.6) reduces to

$$\bar{C}_{\mathrm{GOR}}^{(-\pi/2)} \approx (\alpha_{\mathrm{s}}\Delta z_{\mathrm{s}} - \alpha\Delta z)(\Delta\theta_{\mathrm{s}} - \Delta\theta), \tag{3.9}$$

for a choice of relative phase equivalent to $\psi = -\pi/2$. If the intrinsic optical properties of the sample are homogeneous ($\alpha_{\mathrm{s}} = \alpha$ and $\gamma_{\mathrm{s}} = \gamma$), equation (3.9) is

$$\bar{C}_{\mathrm{GOR}}^{(-\pi/2)} \approx \alpha\gamma(\Delta z_{\mathrm{s}} - \Delta z)^2. \tag{3.10}$$

According to equations (3.5) and (3.10), enantiomorphic samples (for which $\gamma = \pm|\gamma|$) appear with opposite contrasts ($\bar{C}_{\mathrm{GOR}}^{(-\pi/2)} = \pm\,\mathrm{sgn}\alpha|\bar{C}_{\mathrm{GOR}}^{(-\pi/2)}|$). If $|\alpha_{\mathrm{s}}\Delta z_{\mathrm{s}} - \alpha\Delta z| \lesssim 1$ is not satisfied, appropriate compensations might be made by deforming the phase fronts of the sampling and/or reference fields using adaptive optics [33,34].

Interestingly, the initial optical field here (which sports one-dimensional helicity fringes) is essentially the same as the optical field proposed by one of the authors for exerting discriminatory optical forces on chiral molecules [35]; a phenomenon recently demonstrated in the laboratory for chiral liquid crystal microspheres [36]. It is also essentially the same as the optical field employed in the new technique of snapshot circular dichroism, which might enable faster measurements of circular dichroism than is possible using traditional circular dichroism spectrometers [37].

# 4. Simulated results

In this section, we present simulated results for ICOA-OR and ICOA-GOR and compare them with simulated results for other techniques, the latter having been produced according to the theories presented in appendix A. We focus on $\alpha$-quartz, which is uniaxial and chiral; it can adopt either a left-handed ($\mathcal{L}$) or a right-handed ($\mathcal{R}$) crystal structure, which are distinct mirror-image versions of each other. According to one source: '$\alpha$-quartz … is the only optically active and birefringent crystal that has been fully investigated [experimentally]' [8]. Quartz is the second most abundant mineral in the Earth's continental crust [38]. Many gemstones are varieties of quartz [39]. Quartz took centre stage in several seminal studies of optical activity and chirality, including the very discovery of optical activity and the recognition that enantiomorphic forms are associated with opposite signs of optical rotation [2,5,7,12]. Today, it finds use in a variety of applications, ranging from wave plates to digital clocks.

## 4.1. Optical properties of $\alpha$-quartz

In this subsection, we summarize the relevant optical properties of $\alpha$-quartz. We consider light of angular frequency $\omega = 3.7\,\mathrm{Prad\,s^{-1}}$ (green) propagating in the $z$-direction through $\alpha$-quartz with its optic axis

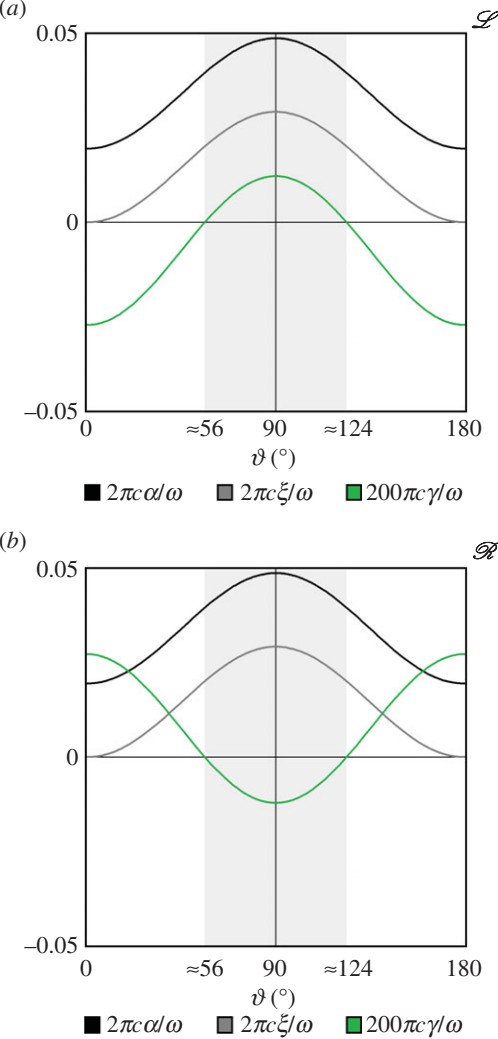

**Figure 3.** (a,b) The intrinsic optical properties $\alpha$, $\xi$ and $\gamma$ as a function of the polar angle $\vartheta$ for both $\mathscr{L}$ structure and $\mathscr{R}$ structure, calculated according to the theory and experimental data described in [6,8,31]. For the sake of clarity, we have amplified the curves pertaining to $\gamma$ by a factor of 100 relative to those pertaining to $\alpha$ and $\xi$.

oriented with azimuthal angle $\varphi$ and polar angle $\vartheta$ in spherical coordinates, taking the $\alpha$-quartz to be immersed in tetralin ($n = 1.541$) for the purpose of refractive index matching [40].

The intrinsic optical properties $\alpha$ and $\gamma$ are independent of the azimuthal angle $\varphi$ whereas the intrinsic optical properties $\beta$ and $\delta$ can be expressed in the form [31]

$$\beta = \xi \sin (2\varphi) \tag{4.1}$$

and

$$\delta = \xi \cos (2\varphi), \tag{4.2}$$

where $\xi = \xi(\vartheta)$ contains all dependence on the polar angle $\vartheta$.

Shown in figure 3 are the intrinsic optical properties $\alpha$, $\xi$ and $\gamma$ as a function of the polar angle $\vartheta$ for both $\mathscr{L}$ structure and $\mathscr{R}$ structure, calculated according to the theory and experimental data described in [6,8,31].

— The mean refractive index parameter $\alpha$ is the same for $\mathscr{L}$ structure and $\mathscr{R}$ structure. It is positive for all values of $\vartheta$ (given our choice of tetralin as a surrounding medium), having its lowest value for light propagating along the optic axis ($\vartheta = 0$ or $\vartheta = 180°$) and its largest value for light propagating perpendicular to the optic axis ($\vartheta = 90°$).

— The linear birefringence parameter $\xi$ is also the same for $\mathscr{L}$ structure and $\mathscr{R}$ structure. It is zero or positive for all values of $\vartheta$, vanishing for light propagating along the optic axis ($\vartheta = 0$ or $\vartheta = 180°$) and having its largest value for light propagating perpendicular to the optic axis ($\vartheta = 90°$).

— The circular birefringence parameter $\gamma$ and thus the optical rotation $\Delta\theta(=\gamma\Delta z)$ of a sample has opposite signs for $\mathscr{L}$ structure and $\mathscr{R}$ structure, thus serving as a signature of chirality. For $\mathscr{L}$ structure, say, it is strongly negative if $0 \le \vartheta < \vartheta_0$ or $180° - \vartheta_0 < \vartheta \le 180°$, vanishing if $\vartheta = \vartheta_0$ or $\vartheta = 180 - \vartheta_0$ and weakly positive if $\vartheta_0 < \vartheta < 180° - \vartheta_0$, where $\vartheta_0 \approx 56°$.

Let us emphasize here that the absolute sign of $\gamma$ and thus $\Delta\theta$ depends on both the chirality *and* orientation of a sample. For $56° \lesssim \vartheta \lesssim 124°$ (the shaded regions in figure 3), $\Delta\theta$ has the opposite sign relative to the case where light propagates along the optic axis ($\vartheta = 0$ or $\vartheta = 180°$); we say that the sample appears 'chiroptically inverted'. These dependences can be traced to the familiar optical rotation tensor $g_{AB}$ ($g_{XX} = g_{YY} = \pm 5.82 \times 10^{-5}$ and $g_{ZZ} = \mp 1.296 \times 10^{-4}$, where the upper signs correspond to $\mathscr{L}$ structure and the lower signs correspond to $\mathscr{R}$ structure); we take

$$\gamma = \frac{\omega}{2c}\left[n_e^{(G)} - n_o^{(G)}\right]\sin\left(\tan^{-1}\left\{\frac{G}{\left[n_e^{(0)} - n_o^{(0)}\right]\sqrt{n_e^{(0)}n_o^{(0)}}}\right\}\right), \tag{4.3}$$

with

$$n_e^{(G)} = \frac{1}{\sqrt{2}}\sqrt{n_e^{(0)2} + n_o^{(0)2} + \sqrt{\left[n_e^{(0)2} - n_o^{(0)2}\right]^2 + 4G^2}} \tag{4.4}$$

and

$$n_o^{(G)} = \frac{1}{\sqrt{2}}\sqrt{n_e^{(0)2} + n_o^{(0)2} - \sqrt{\left[n_e^{(0)2} - n_o^{(0)2}\right]^2 + 4G^2}} \tag{4.5}$$

using

$$G = g_{XX}\sin^2\vartheta + g_{ZZ}\cos^2\vartheta \tag{4.6}$$

$$n_e^{(0)} = \frac{n_e n_o}{\sqrt{n_e^2\cos^2\vartheta + n_o^2\sin^2\vartheta}} \tag{4.7}$$

and
$$n_o^{(0)} = n_o, \tag{4.8}$$

where $n_e^{(G)}$ is the extraordinary refractive index and $n_o^{(G)}$ is the ordinary refractive index.

## 4.2. Brazil twinning

In this subsection, we consider the phenomenon of Brazil twinning. According to one source: 'A Brazil twin in quartz consists of a left-handed region and a right-handed region in contact.' [41].

### 4.2.1. Samples I, II and II (tilted)

Depicted in figure 4 are three illustrative samples, each consisting of a basal slice of $\alpha$-quartz in which Brazil twinning occurs.[1] The slices are immersed in tetralin for the purpose of refractive index matching and are optically flat ($|\Delta(\alpha\Delta z)| \ll 1$ across the slices). Sample I has a single planar Brazil-twin boundary and is oriented with its optic axis parallel to the direction of propagation of the light; see panel (*a*) Sample II has two intersecting planar Brazil-twin boundaries and is also oriented with its optic axis parallel to the direction of propagation of the light; see panel (*b*). Sample II (tilted) is obtained from sample II by rotating the slice first through 15° about the *x*-axis then through 15° about the *z* axis; see panel (*c*).

---

[1]To obtain the Jones matrix $\bar{M}$ of a slice we decomposed the slice into segments of uniform structure ($\mathscr{L}$ or $\mathscr{R}$) and multiplied the Jones matrices of each of these segments together in the appropriate order. Interestingly, sample II (tilted) has (small) 'anomalous' contributions to the circular birefringence parameter $\gamma$ along its twin boundaries, these being mathematically attributable to the fact that the Jones matrices of adjacent segments do not commute [31].

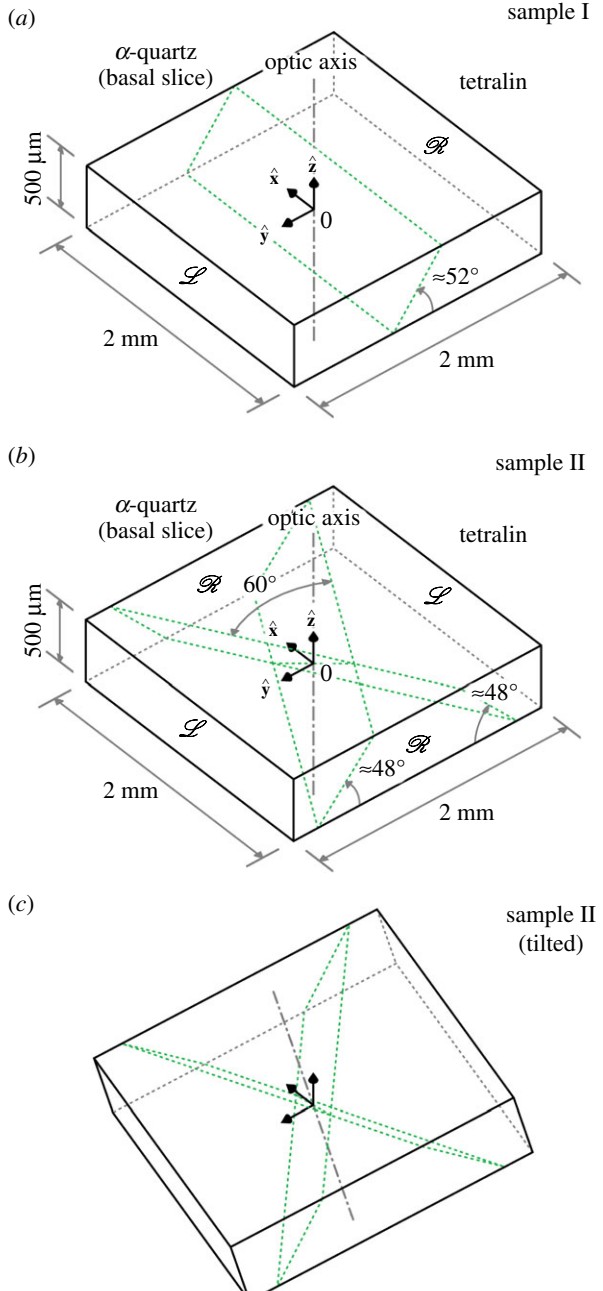

**Figure 4.** (a–c) Three illustrative samples, each consisting of a basal slice of α-quartz in which Brazil twinning occurs.

### 4.2.2. Appearance using 'crossed' polarizers

Shown in figure 5 are simulated images of sample I, sample II and sample II (tilted) as they would appear using 'crossed' polarizers, according to equation (A 4).

In panels (a) and (b), regions of $\mathscr{L}$ structure appear darkened and regions of $\mathscr{R}$ structure appear brightened, with contrasts of equal magnitude; for the special case of observation along the optic axis, α-quartz exhibits zero linear birefringence and the relationship between optical rotation and the contrast obtained using 'crossed' polarizers is simple. The contrast is unchanged if sample I or sample II is rotated about the direction of propagation of the light.

In panel (c), however, regions of $\mathscr{L}$ structure and regions of $\mathscr{R}$ structure both appear brightened, with contrasts of unequal magnitude; for observation oblique to the optic axis, α-quartz exhibits non-zero linear birefringence and the relationship between optical rotation and the contrast obtained using 'crossed' polarizers is complicated in general. The contrast changes if sample II (tilted) is rotated about the direction of propagation of the light.

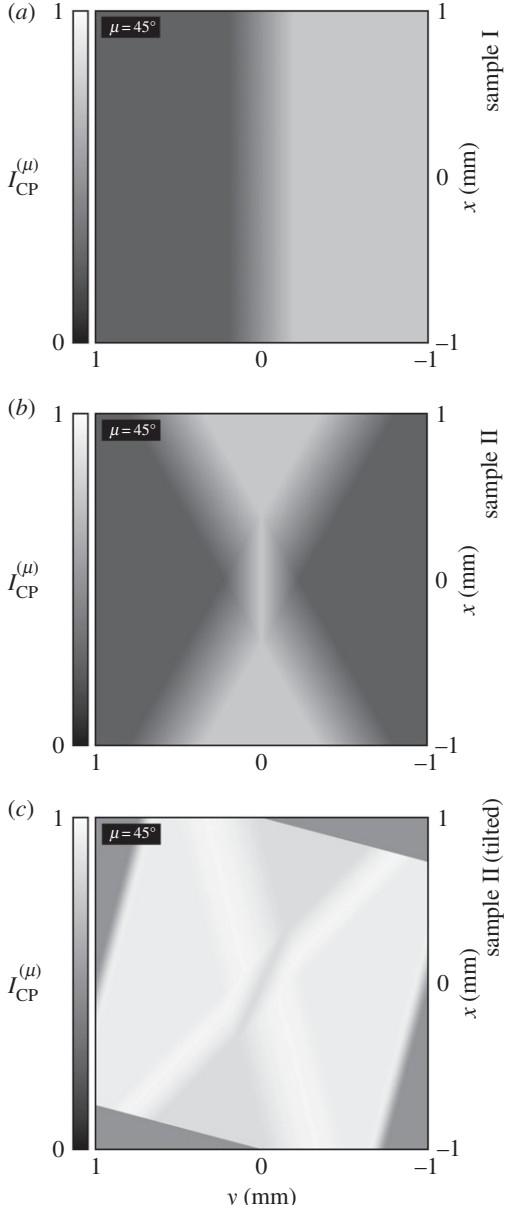

**Figure 5.** (a–c) Simulated images of sample I, sample II and sample II (tilted) as they would appear using 'crossed' polarizers, according to equation (A 4).

Note that 'crossed' polarizers can yield non-zero contrast for a transparent sample exhibiting non-zero linear birefringence even if the optical rotation of the sample is zero; they are not particularly well suited to the study of optical rotation.

### 4.2.3. Appearance using ICOA-OR

Shown in figure 6 are simulated images of sample I, sample II and sample II (tilted) as they would appear using ICOA-OR, according to equations (2.7) and (2.8).

In panels (a), (b) and (c), regions of $\mathscr{L}$ structure appear darkened and regions of $\mathscr{R}$ structure appear brightened, with contrasts of equal magnitude; for observation of $\alpha$-quartz along the optic axis *or* oblique to the optic axis, the relationship between optical rotation and the contrast obtained using ICOA-OR is simple (assuming that $|\alpha|\Delta z \lesssim 1$ or $|\Delta(\alpha\Delta z)| \lesssim 1$ and $\tau\Delta z \lesssim 1$). The contrast is unchanged if sample I, sample II *or* sample II (tilted) is rotated about the direction of propagation of the light.

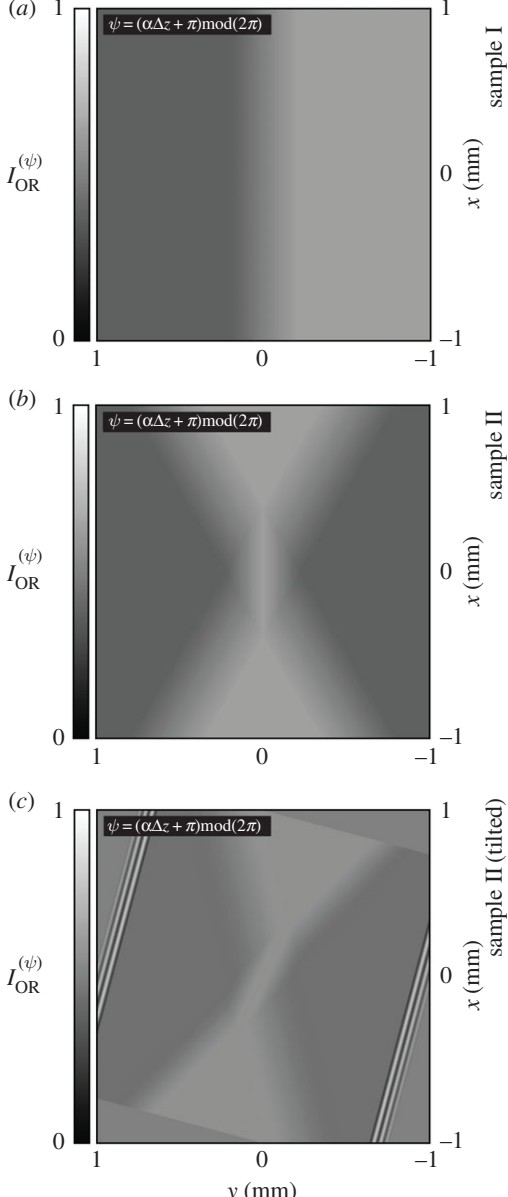

**Figure 6.** (*a–c*) Simulated images of sample I, sample II and sample II (tilted) as they would appear using ICOA-OR, according to equations (2.7) and (2.8).

Let us emphasize here that ICOA-OR can yield a non-zero contrast for a transparent sample if and only if the optical rotation of the sample is non-zero, *regardless* of the linear birefringence of the sample; it enables the detection of optical rotation without ambiguity.

## 4.3. Microscopic crystals and enantiomorphic identification

In this subsection, we consider microscopic crystals and the task of enantiomorphic identification; given a sample of such crystals, we seek to identify which have $\mathscr{L}$ structure and which have $\mathscr{R}$ structure, thus enabling the determination of the enantiomorphic excess

$$EE = \frac{N_{\mathscr{R}} - N_{\mathscr{L}}}{N_{\mathscr{R}} + N_{\mathscr{L}}} \qquad (4.9)$$

of the sample, where $N_{\mathscr{L}}$ is the number of crystals with $\mathscr{L}$ structure and $N_{\mathscr{R}}$ is the number with $\mathscr{R}$ structure.

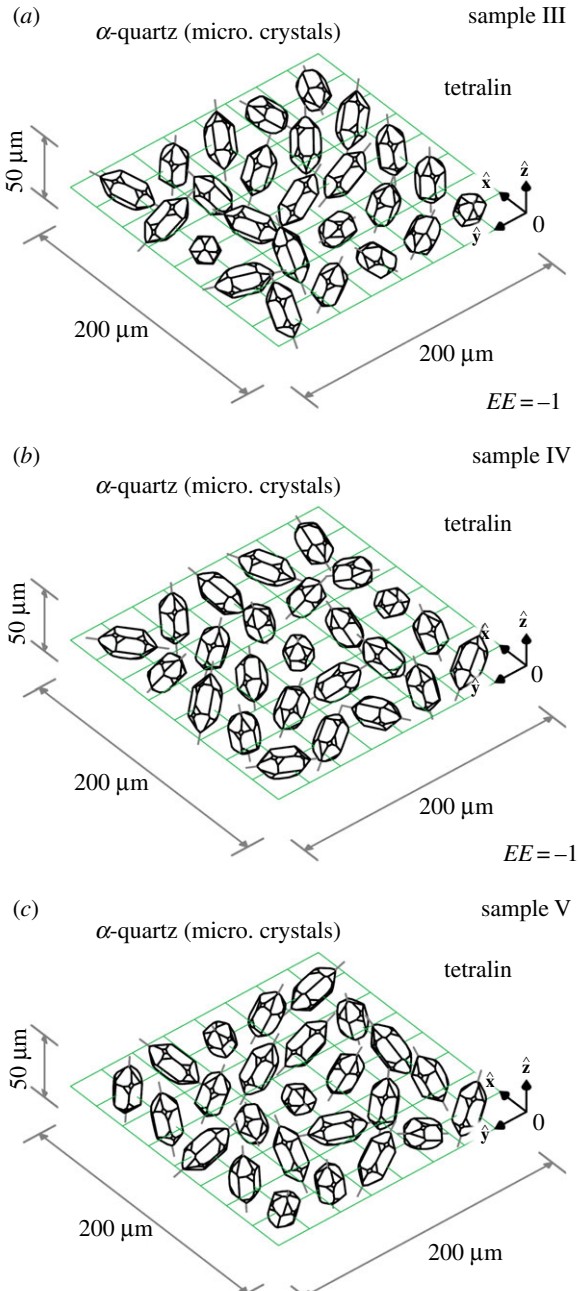

**Figure 7.** ($a$–$c$) Three illustrative samples, each consisting of 25 microscopic hemihedral $\alpha$-quartz crystals with randomly chosen orientations.

### 4.3.1. Samples III, IV and V

Depicted in figure 7 are three illustrative samples, each consisting of twenty-five microscopic hemihedral $\alpha$-quartz crystals with randomly chosen orientations. The crystals are immersed in tetralin for the purpose of refractive index matching and are optically thin ($|\alpha_s \Delta z_s - \alpha \Delta z| \lesssim 1$ everywhere). Sample III is enantiopure, being composed solely of crystals with $\mathscr{L}$ structure ($EE = -1$); see panel ($a$). Sample IV is also enantiopure, being composed solely of crystals with $\mathscr{R}$ structure ($EE = 1$); see panel ($b$). Sample V is scalemic, being composed of a mixture of crystals with $\mathscr{L}$ structure and crystals with $\mathscr{R}$ structure ($EE \neq \pm 1$); see panel ($c$).

### 4.3.2. Appearance using standard DIC

Shown in figure 8 are simulated images of sample III, sample IV and sample V as they would appear using standard DIC, according to equation (A 7).

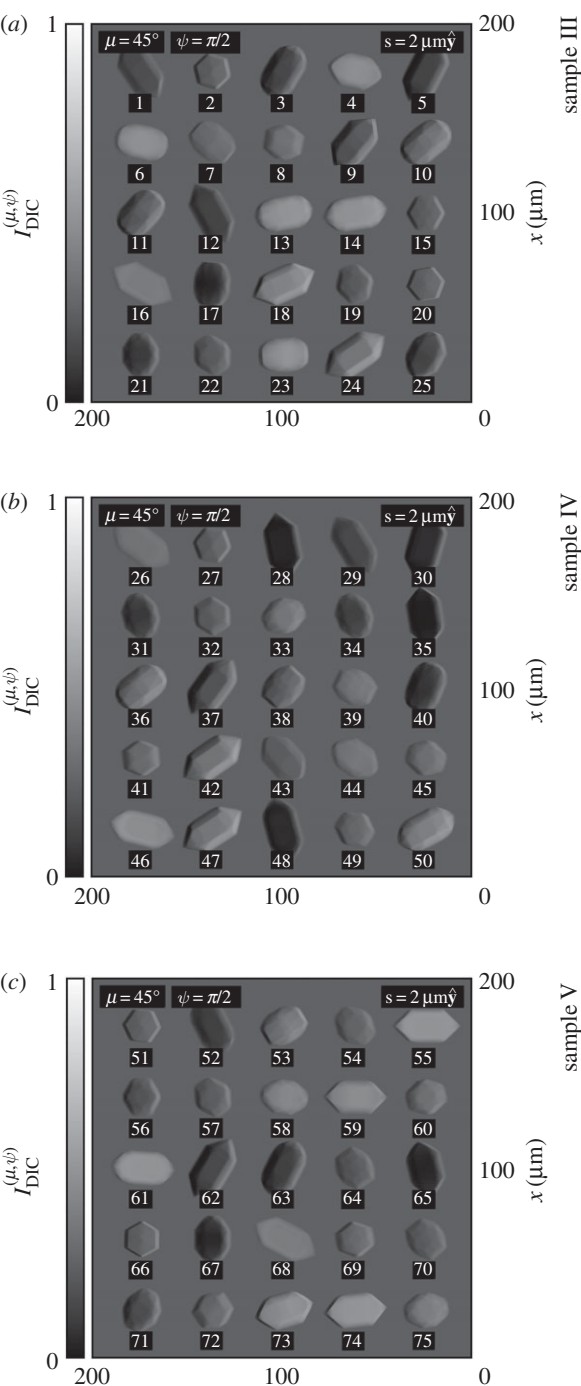

**Figure 8.** (a–c) Simulated images of sample III, sample IV and sample V as they would appear using standard DIC, according to equation (A 7).

In panels (a), (b) and (c), each crystal appears with its left-hand side darkened and its right-hand side brightened, *regardless* of whether the crystal has $\mathscr{L}$ or $\mathscr{R}$ structure; see crystals 1–75.

Standard DIC is chirally insensitive at leading order; it is not particularly well suited to the study of optical rotation.

### 4.3.3. Appearance using ICOA-GOR

Shown in figure 9 are simulated images of sample III, sample IV and sample V as they would appear using ICOA-GOR with a choice of relative phase equivalent to either $\psi = 0$ or $\psi = -\pi/2$, according to equations (3.5) and (3.6). For each crystal, the sign of the optical rotation $\Delta\theta$ can be seen immediately.

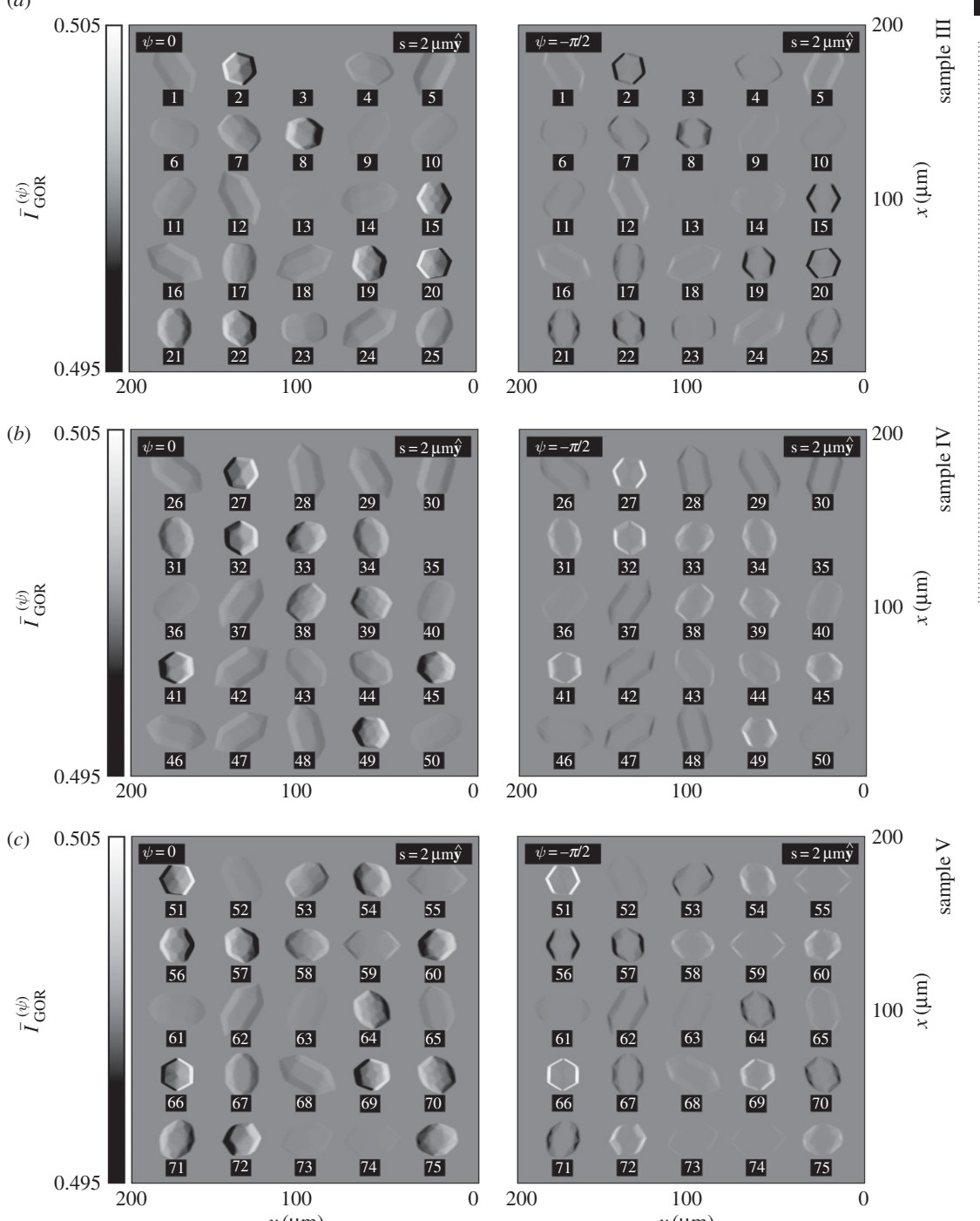

**Figure 9.** (*a*–*c*) Simulated images of sample III, sample IV and sample V as they would appear using ICOA-GOR with a choice of relative phase equivalent to either $\psi = 0$ or $\psi = -\pi/2$, according to equations (3.5) and (3.6). For the sake of clarity, we have plotted the intensity $\bar{I}_{GOR}^{(\psi)}$ for values in the range [0.495, 0.505] rather than [0, 1]; black and white correspond to contrasts of $\bar{C}_{GOR}^{(\psi)} = \mp 1\%$.

Together with knowledge about the orientation of the crystal (in particular, whether the crystal appears 'chiroptically inverted' or not), this can be used to determine the structure ($\mathscr{L}$ or $\mathscr{R}$) of the crystal.

Panel (*a*) illustrates three distinct possibilities for a crystal with $\mathscr{L}$ structure. If the optic axis of the crystal makes an angle of less than $\approx 56°$ with the direction of propagation of the light, the crystal appears with its left-hand side strongly brightened and its right-hand side strongly darkened for $\psi = 0$ or simply with both sides strongly darkened for $\psi = -\pi/2$ (the optical rotation is strongly negative); see crystals 2, 4, 6–8, 10, 11, 15, 17, 19–23 and 25. If the optic axis makes an angle of around $\approx 56°$, the crystal is essentially invisible (the optical rotation is essentially zero); see crystals 3 and 13. If the optic axis makes angles greater than $\approx$

56°, the crystal appears with its right-hand side weakly brightened and its left-hand weakly darkened for $\psi = 0$ or simply with both sides weakly brightened for $\psi = -\pi/2$ (the optical rotation is weakly positive, as the crystal appears 'chiroptically inverted'); see crystals 1, 5, 9, 12, 14, 16, 18 and 24.

Panel (*b*) illustrates three *complementary* possibilities for a crystal with $\mathscr{R}$ structure. If the optic axis of the crystal makes an angle of less than ≈56° with the direction of propagation of the light, the crystal appears with its right-hand side strongly brightened and its left-hand side strongly darkened for $\psi = 0$ or simply with both sides strongly brightened for $\psi = -\pi/2$ (the optical rotation is strongly positive); see crystals 27, 31–34, 38–41, 43–45 and 49. If the optic axis makes an angle of around ≈56°, the crystal is essentially invisible (the optical rotation is essentially zero); see crystal 35. If the optic axis makes angles greater than ≈56°, the crystal appears with its left-hand side weakly brightened and its right-hand weakly darkened for $\psi = 0$ or simply with both sides weakly darkened for $\psi = -\pi/2$ (the optical rotation is weakly negative, as the crystal appears 'chiroptically inverted'); see crystals 26, 28–30, 37, 42, 46–48 and 50.

The reader is invited to try their hand at the task of enantiomorphic identification and thus the determination of the enantiomorphic excess *EE* of sample V using panel (*c*) together with panels (*a*) and (*b*) for guidance.[2] The optic axes of crystals 51, 53, 54, 56–60, 64–67, 69–73 and 75 each make an angle of less than ≈56° with the direction of propagation of the light, and those of crystals 52, 55, 61–63, 68 and 74 each make angles of more than ≈56°; the latter crystals appear 'chiroptically inverted'.

It has recently been shown that PlasDIC can be applied to help monitor crystal growth, with potential applications in the pharmaceutical industry [26,28]. The results presented above lead us to suggest that ICOA-GOR might be applied to provide chiral sensitivity in this context, an important addition given the importance of chirality to the pharmaceutical industry [42]. We envisage using ICOA-GOR to determine the sign of the optical rotation of a crystal in conjunction with a computer algorithm trained with knowledge of the crystal's habit to determine the crystal's orientation; together, this information will reveal the chirality of the crystal.

# 5. Outlook

Let us conclude by highlighting some possible directions for future research.

— The Jones vector formalism neglects numerous effects; diffraction, depolarization and more [2,22,31,32]. It is desirable to develop a more accurate theoretical description of ICOA.
— Closely related is the task of modelling imperfections in real 'chiral microscopes' based on ICOA.
— It remains for us to elucidate the use of ICOA for probing manifestations of optical activity other than optical rotation; the circular dichroism of absorbing samples, for example, as well as versions of ICOA based on reflection rather than transmission.
— There are a wealth of potential applications to be explored for ICOA and associated 'chiral microscopes'.

We will return to these and related tasks elsewhere.

Ethics. The authors declare that this manuscript conforms to the ethical code of the Royal Society.
Data accessibility. This manuscript has no additional data.
Authors' contributions. U.V. and N.T. initiated this work by asking whether a chiral variant of DIC is possible; R.P.C. conceived of ICOA and performed the necessary calculations. All authors gave final approval for publication.
Competing interests. The authors are named inventors in a German patent application (DE 10 2019 117 671.9) based on ICOA.
Funding. This work was supported by the Leverhulme Trust grant no. (RPG-2017-048) and the Royal Society (URF/R1/191243). R.P.C. is a Royal Society University Research Fellow.
Acknowledgements. The authors gratefully acknowledge helpful discussions with Holger Münz.

# Appendix A. Other techniques

In this appendix, we present theories describing the use of 'crossed' polarizers and standard DIC.

## A.1. 'Crossed' polarizers

In this subsection, we consider the use of 'crossed' polarizers. For the sake of concreteness, we consider a basic set-up similar to that described in §2 for ICOA-OR but with the polarization rotators $\mathcal{PR}_1$ and $\mathcal{PR}_2$

---

[2]Crystals 53, 55–57, 64, 67, 68, 70, 71 and 74 have $\mathscr{L}$ structure and crystals 51, 52, 54, 58–63, 65, 66, 69, 72, 73, and 75 have $\mathscr{R}$ structure; the enantiomorphic excess of sample V is $EE = (15 - 10)/(15 + 10) = 1/5$.

switched off ($\sigma = 0$), the reference field blocked ($\tilde{J}_r^{(0)} \to 0$) and beam splitter $\mathcal{BS}_2$ replaced by an analyser, the latter being described by the Jones matrix

$$A^{(\mu)} = \begin{bmatrix} \cos^2 \mu & -\sin \mu \cos \mu \\ -\sin \mu \cos \mu & \sin^2 \mu \end{bmatrix},$$ (A 1)

where $\mu$ is dictated by the orientation of the analyser. The Jones vector of the final field is thus

$$\tilde{J}_f^{(\mu)} = A^{(\mu)} \tilde{M} \tilde{J}_s^{(0)},$$ (A 2)

and the intensity of the final field follows as:

$$\begin{aligned} I_{CP}^{(\mu)} &= 2 \tilde{J}_f^{(\mu)\dagger} \tilde{J}_f^{(\mu)} \\ &= \cos^2 \mu \cos^2 (\tau \Delta z) + \left[ \cos^2 \mu \delta^2 \Delta z^2 + \sin^2 \mu (\beta^2 \Delta z^2 + \Delta \theta^2) \right] \mathrm{sinc}^2(\tau \Delta z) \\ &\quad + \sin(2\mu) \left[ \Delta \theta \cos(\tau \Delta z) - \beta \Delta z \delta \Delta z \, \mathrm{sinc}(\tau \Delta z) \right] \mathrm{sinc}(\tau \Delta z), \end{aligned}$$ (A 3)

normalized here to give values in the range [0, 1]. Equation (A 3) reduces to

$$I_{CP}^{(\pm 45^\circ)} = \frac{1}{2} \pm \left[ \Delta \theta \cos(\tau \Delta z) - \beta \Delta z \delta \Delta z \, \mathrm{sinc}(\tau \Delta z) \right] \mathrm{sinc}(\tau \Delta z)$$ (A 4)

for a choice of analyser orientation equivalent to $\mu = \pm 45^\circ$.

## A.2. Standard DIC

In this subsection, we consider standard DIC. For the sake of concreteness we consider a basic set-up similar to that described in §3 for ICOA-GOR but with the polarization rotator $\mathcal{PR}$ switched off ($\sigma = 0$) and an analyser placed before the detector $\mathcal{D}$, the analyser being described by the Jones matrix

$$A^{(\pm 45^\circ)} = \frac{1}{2} \begin{bmatrix} 1 & \mp 1 \\ \mp 1 & 1 \end{bmatrix},$$ (A 5)

as follows from equation (A 1) for a choice of analyser orientation equivalent to $\mu = \pm 45^\circ$. The Jones vector of the final field at the position $z = z_\mathcal{D}$ of $\mathcal{D}$ is thus

$$\tilde{J}_f^{(\pm 45^\circ, \psi)}(z = z_\mathcal{D}) = A^{(\pm 45^\circ)} \left[ \tilde{M} \tilde{J}_s^{(0)} + e^{i\psi} \tilde{M}_s \tilde{J}_r^{(0)} \right],$$ (A 6)

and the intensity of the final field follows as:

$$\begin{aligned} I_{DIC}^{(\pm 45^\circ, \psi)} &= \frac{1}{2} \tilde{J}_f^{(\pm 45^\circ, \psi)\dagger}(z = z_\mathcal{D}) \tilde{J}_f^{(\pm 45^\circ, \psi)}(z = z_\mathcal{D}) \\ &= \frac{1}{8} \left[ \cos^2 (\tau_s \Delta z_s) \mp 2 \Delta \theta_s \cos(\tau_s \Delta z_s) \, \mathrm{sinc}(\tau_s \Delta z_s) \right. \\ &\quad \left. + (\beta_s \Delta z_s - i \Delta \theta_s \pm \delta_s \Delta z_s)(\beta_s \Delta z_s + i \Delta \theta_s \pm \delta_s \Delta z_s) \mathrm{sinc}^2(\tau_s \Delta z_s) \right] \\ &\quad + \frac{1}{8} \left[ \cos^2 (\tau \Delta z) \pm 2 \Delta \theta \cos(\tau \Delta z) \, \mathrm{sinc}(\tau \Delta z) \right. \\ &\quad \left. + (\beta \Delta z - i \Delta \theta \mp \delta \Delta z)(\beta \Delta z + i \Delta \theta \mp \delta \Delta z) \mathrm{sinc}^2(\tau \Delta z) \right] \\ &\quad \mp \frac{1}{4} \Re \left\{ e^{i(\alpha_s \Delta z_s - \alpha \Delta z + \psi)} \left[ \cos(\tau_s \Delta z_s) \mp i(\beta_s \Delta z_s - i \Delta \theta_s \pm \delta_s \Delta z_s) \, \mathrm{sinc}(\tau_s \Delta z_s) \right] \right. \\ &\quad \left. \times \left[ \cos(\tau \Delta z) \pm i(\beta \Delta z - i \Delta \theta \mp \delta \Delta z) \, \mathrm{sinc}(\tau \Delta z) \right] \right\}, \end{aligned}$$ (A 7)

normalized here to give values in the range [0, 1]. Equation (A 7) reduces to

$$I_{DIC}^{(\pm 45^\circ, \psi)} = \frac{1}{4} \left[ 1 \mp \cos(\alpha_s \Delta z_s - \alpha \Delta z + \psi) \right],$$ (A 8)

for $\beta_s = \gamma_s = \delta_s = \beta = \gamma = \delta = 0$. Equation (A 8) is the result usually quoted in simple descriptions of standard DIC.

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
