## [Reviewer comments · Royal Society Open Science]

Review History

RSOS-192201.R0 (Original submission)

Review form: Reviewer 1 (Nirmal Mazumder)

Is the manuscript scientifically sound in its present form?

Yes

Are the interpretations and conclusions justified by the results?

Yes

Is the language acceptable?

Yes

Do you have any ethical concerns with this paper?

No

Have you any concerns about statistical analyses in this paper?

No

Recommendation?

Accept with minor revision (please list in comments)

Comments to the Author(s)

The authors describes interference-contrast optical activity (ICOA) and try to probe chirality. The article can be accepted with minor revision.

The author mentioned ICOA for probing the chirality of anisotropic samples. I feel that this is kind of confusion.

All chiral molecules are not anisotropic. author should clarify it.

Author should comment on polarization microscopy. is ICOA is different than polarization microscopy.

Jones and Stokes vectors are used for finding the anisotropy and chirality of the molecules. is ICOA is better than that ?

I feel that the experimental results may depend on wavelength of light source. will it affect the interpretation ?

Review form: Reviewer 2 (Oriol Arteaga)

Is the manuscript scientifically sound in its present form?

Yes

Are the interpretations and conclusions justified by the results?

Yes

Is the language acceptable?

Yes

Do you have any ethical concerns with this paper?

Yes

Have you any concerns about statistical analyses in this paper?

No

Recommendation?

Major revision is needed (please make suggestions in comments)

Comments to the Author(s)

See attached (Appendix A).

Decision letter (RSOS-192201.R0)

27-Feb-2020

Dear Dr Cameron,

The editors assigned to your paper ("Interference-contrast optical activity (ICOA): a new technique for probing the chirality of anisotropic samples and more") have now received

comments from reviewers. We would like you to revise your paper in accordance with the referee and Associate Editor suggestions which can be found below (not including confidential reports to the Editor). Please note this decision does not guarantee eventual acceptance.

Please submit a copy of your revised paper before 21-Mar-2020. Please note that the revision deadline will expire at 00.00am on this date. If we do not hear from you within this time then it will be assumed that the paper has been withdrawn. In exceptional circumstances, extensions may be possible if agreed with the Editorial Office in advance. We do not allow multiple rounds of revision so we urge you to make every effort to fully address all of the comments at this stage. If deemed necessary by the Editors, your manuscript will be sent back to one or more of the original reviewers for assessment. If the original reviewers are not available, we may invite new reviewers.

- Data accessibility

<http://datadryad.org/submit?journalID=RSOS&manu=RSOS-192201>

- Competing interests

- Authors' contributions

All submissions, other than those with a single author, must include an Authors' Contributions section which individually lists the specific contribution of each author. The list of Authors

should meet all of the following criteria; 1) substantial contributions to conception and design, or acquisition of data, or analysis and interpretation of data; 2) drafting the article or revising it critically for important intellectual content; and 3) final approval of the version to be published.

- Acknowledgements

- Funding statement

on behalf of Dr Peter Munro (Associate Editor) and Miles Padgett (Subject Editor)
openscience@royalsociety.org

Reviewers' Comments to Author:

Reviewer: 1

Comments to the Author(s)

The authors describes interference-contrast optical activity (ICOA) and try to probe chirality. The article can be accepted with minor revision.

The author mentioned ICOA for probing the chirality of anisotropic samples. I feel that this is kind of confusion.

All chiral molecules are not anisotropic. author should clarify it.

Author should comment on polarization microscopy. is ICOA is different than polarization microscopy.

Jones and Stokes vectors are used for finding the anisotropy and chirality of the molecules. is ICOA is better than that ?

I feel that the experimental results may depend on wavelength of light source. will it affect the interpretation ?

Reviewer: 2

Comments to the Author(s)
See attached ("Cameron.pdf")

Author's Response to Decision Letter for (RSOS-192201.R0)

See Appendix B.

RSOS-192201.R1 (Revision)

Review form: Reviewer 1 (Nirmal Mazumder)

Is the manuscript scientifically sound in its present form?

Yes

Are the interpretations and conclusions justified by the results?

Yes

Is the language acceptable?

Yes

Do you have any ethical concerns with this paper?

Yes

Have you any concerns about statistical analyses in this paper?

Yes

Recommendation?

Accept as is

Comments to the Author(s)

The manuscript is accepted in present form.

Review form: Reviewer 2 (Oriol Arteaga)

Is the manuscript scientifically sound in its present form?

Yes

Are the interpretations and conclusions justified by the results?

Yes

Is the language acceptable?

Yes

Do you have any ethical concerns with this paper?

No

Have you any concerns about statistical analyses in this paper?

No

Recommendation?

Accept with minor revision (please list in comments)

Comments to the Author(s)

Authors have done a good revision of the manuscript and they have answered clearly to the questions raised by the reviewers. I specially appreciate the more clear explanation of the ICOA-GOR method. I think the paper can be accepted for publication.

A couple of minor additional questions I have (authors may optionally want to consider them):

- The lens free method proposed for ICOA-GOR seems to have not very large lateral resolution and this can be contradictory with the examples of microscopic quartz crystals provided.

- In the context of ICOA-GOR authors should probably cite other papers in which interferometric single-beam setups have been used to measure optical activity (e.g. 10.1002/mop.4650060706 and 10.1364/OE.27.006746)

Decision letter (RSOS-192201.R1)

07-Apr-2020

Dear Dr Cameron,

On behalf of the Editors, I am pleased to inform you that your Manuscript RSOS-192201.R1 entitled "Interference-contrast optical activity (ICOA): a new technique for probing the chirality of anisotropic samples and more" has been accepted for publication in Royal Society Open Science subject to minor revision in accordance with the referee suggestions. Please find the referees' comments at the end of this email.

The reviewers and Subject Editor have recommended publication, but also suggest some minor revisions to your manuscript. Therefore, I invite you to respond to the comments and revise your manuscript.

- **Ethics statement**

- **Data accessibility**

It is a condition of publication that all supporting data are made available either as supplementary information or preferably in a suitable permanent repository. The data accessibility section should state where the article's supporting data can be accessed. This section should also include details, where possible of where to access other relevant research materials such as statistical tools, protocols, software etc can be accessed. If the data has been deposited in an external repository this section should list the database, accession number and link to the DOI

for all data from the article that has been made publicly available. Data sets that have been deposited in an external repository and have a DOI should also be appropriately cited in the manuscript and included in the reference list.

<http://datadryad.org/submit?journalID=RSOS&manu=RSOS-192201.R1>

- **Competing interests**

- **Authors' contributions**

- **Acknowledgements**

- **Funding statement**

Because the schedule for publication is very tight, it is a condition of publication that you submit the revised version of your manuscript before 16-Apr-2020. Please note that the revision deadline will expire at 00.00am on this date. If you do not think you will be able to meet this date please let me know immediately.

When submitting your revised manuscript, you will be able to respond to the comments made by the referees and upload a file "Response to Referees" in "Section 6 - File Upload". You can use this to document any changes you make to the original manuscript. In order to expedite the

processing of the revised manuscript, please be as specific as possible in your response to the referees.

Kind regards,
Lianne Parkhouse
Editorial Coordinator
Royal Society Open Science
openscience@royalsociety.org

on behalf of Dr Peter Munro (Associate Editor) and Miles Padgett (Subject Editor)
openscience@royalsociety.org

Associate Editor Comments to Author (Dr Peter Munro):

Please consider making the minor revisions suggested by the second reviewer.

Reviewer comments to Author:

Reviewer: 1
Comments to the Author(s)

The manuscript is accepted in present form.

Reviewer: 2
Comments to the Author(s)

Authors have done a good revision of the manuscript and they have answered clearly to the questions raised by the reviewers. I specially appreciate the more clear explanation of the ICOA-GOR method. I think the paper can be accepted for publication.

A couple of minor additional questions I have (authors may optionally want to consider them):

- The lens free method proposed for ICOA-GOR seems to have not very large lateral resolution and this can be contradictory with the examples of microscopic quartz crystals provided.
- In the context of ICOA-GOR authors should probably cite other papers in which interferometric single-beam setups have been used to measure optical activity (e.g. 10.1002/mop.4650060706 and 10.1364/OE.27.006746)

Author's Response to Decision Letter for (RSOS-192201.R1)

See Appendix C.

Decision letter (RSOS-192201.R2)

17-Apr-2020

Dear Dr Cameron,

It is a pleasure to accept your manuscript entitled "Interference-contrast optical activity (ICOA): a new technique for probing the chirality of anisotropic samples and more" in its current form for publication in Royal Society Open Science.

on behalf of Dr Peter Munro (Associate Editor) and Miles Padgett (Subject Editor)
openscience@royalsociety.org

Appendix A

The paper by R. P. Cameron, U. Vogl and N. Trautmann proposes a new technique based on polarization interferometry to quantify the circular birefringence (optical rotation) in anisotropic samples. The work suggests that a new type of interference contrast microscopes (“chiral microscopes”) could use this technique for example to monitor the growth of chiral crystals.

The idea that the paper explores, an interference contrast that depends only on the optical activity and not on the linear birefringence, is very interesting and it is worth considering. The manuscript is very well written and the ideas are clearly presented. The only (major) problem that I find is that the real feasibility of the technique is not discussed. The entire paper is based on simulations and assuming ideal optical components and detectors, but being optical activity on anisotropic materials “very hard to measure” it is desirable that any proposed technique does not disregard the experimental feasibility. I think that there are two main points that need discussion:

- The contrast shown in Fig. 8 is really small. Note that all the grayscale levels are contained in just 0.01. Can imaging detectors detect this very small interferometric visibility in real experimental conditions of noise?
- The basic setup in Fig. 1 contains four optical components (two mirrors and two beam splitters) which will introduce changes in the polarization of the beams that will be at least one order of magnitude larger than the optical rotation caused by a microscopic crystal. This is a particularly bad situation for beamsplitters where even does marketed “non-polarizing” have a rather large effect on polarization. Therefore it is expected that final interference contrast images can have very significant artifacts that, most likely, will be not treatable as systematic errors as they will also depend on the linear birefringence of the sample.

Minor issues

- In Fig. 2 the line breaking between “100” and “ $\times 2 \cdot \pi \cdot c \dots$ ” can be confusing. It should be clearer in the legend and also in the caption that the green line has been multiplied by a factor 100.
- For a better understanding of section 4a and Fig.2 reference should be made to the opposite signs of the optical activity tensors components of quartz, which produces different optical rotation signs depending on the angle under consideration and zero optical rotation at a specific angle.

Appendix B

Response to Referees

for “Interference-contrast optical activity (ICOA): a new technique for probing the chirality of anisotropic samples and more” (RSOS-192201)

Reproduced below is the initial decision letter from **Royal Society Open Science**, the initial report from **Referee 1** and the initial report from **Referee 2**, together with our responses to the referees.

Initial decision letter from **Royal Society Open Science**:

Dear Dr Cameron,

The editors assigned to your paper ("Interference-contrast optical activity (ICOA): a new technique for probing the chirality of anisotropic samples and more") have now received comments from reviewers. We would like you to revise your paper in accordance with the referee and Associate Editor suggestions which can be found below (not including confidential reports to the Editor). Please note this decision does not guarantee eventual acceptance.

Please submit a copy of your revised paper before 21-Mar-2020. Please note that the revision deadline will expire at 00.00am on this date. If we do not hear from you within this time then it will be assumed that the paper has been withdrawn. In exceptional circumstances, extensions may be possible if agreed with the Editorial Office in advance. We do not allow multiple rounds of revision so we urge you to make every effort to fully address all of the comments at this stage. If deemed necessary by the Editors, your manuscript will be sent back to one or more of the original reviewers for assessment. If the original reviewers are not available, we may invite new reviewers.

- Data accessibility

<http://datadryad.org/submit?journalID=RSOS&manu=RSOS-192201>

- Competing interests

- Authors' contributions

- Acknowledgements

- Funding statement

Kind regards,

Anita Kristiansen
Editorial Coordinator

on behalf of Dr Peter Munro (Associate Editor) and Miles Padgett (Subject Editor)
openscience@royalsociety.org

Initial report from **Referee 1** and our responses:

The authors describes interference-contrast optical activity (ICOA) and try to probe chirality. The article can be accepted with minor revision.

We thank **Referee 1** for their recommendation that our manuscript be published, subject to minor revision.

The author mentioned ICOA for probing the chirality of anisotropic samples. I feel that this is kind of confusion.

All chiral molecules are not anisotropic. author should clarify it.

We have not claimed that all chiral molecules (or samples) are anisotropic. In fact, the first sentence of our manuscript acknowledges the opposite, reading as follows:

“Optical rotation, circular dichroism and other manifestations of optical activity are measured routinely for isotropic samples, serving as hallmarks of chirality in applications ranging from the determination of sugar concentrations to the investigation of virus structures \cite{Lough02a, Barron04a, Berova12a, Polavarapu18a}.”

Author should comment on polarization microscopy. is ICOA is different than polarization microscopy.

The second paragraph of our revised manuscript includes the following text:

“ICOA-OR and ICOA-GOR are distinct from all polarimetric techniques known to the authors at the time of writing, including HAUP-based techniques \cite{Kobayashi83a, Kobayashi83b, Kaminsky96a, Kaminsky04a, Claborn06a}, optical heterodyne polarimetry \cite{King93a, Chou97a}, Metripol-based techniques \cite{Glazer96a, Kaminsky04a}, CRDP and other such cavity-based techniques \cite{Muller00a, Muller02a, Sofikitis18a} and Mueller matrix polarimetry \cite{Arteaga10a, Freudenthal12a}. ICOA-GOR in particular has elements in common with, but is subtly distinct from, DIC-based techniques \cite{Smith47a, Nomarski52a, Lang68a, Danz04a, Danz04b, Arnaout16a, Terborg16a}. ICOA is complementary to chiral rotational spectroscopy; a technique proposed recently by one of the authors to determine orientated chiroptical information about individual molecules \cite{Cameron16a}.”

Furthermore, simulated results for ICOA-OR are compared with simulated results for ‘crossed’ polarisers in section 4b).

Jones and Stokes vectors are used for finding the anisotropy and chirality of the molecules. is ICOA is better than that ?

Our manuscript *is* based on a Jones vector analysis. To make this clearer, we have included the following text in section 1 of our revised manuscript:

“We work in the domain of classical optics using the Jones vector formalism \cite{Jones41a, Schellman87a, Barron04a, Arteaga10a}, with the upper components of our Jones vectors corresponding to the E_x component of the electric field and the lower components corresponding to the E_y component.”

I feel that the experimental results may depend on wavelength of light source. will it affect the interpretation ?

Monochromatic light is considered throughout our manuscript, as coherence is necessary to generate the desired interference contrast.

The optical properties of a sample will vary, of course, with the chosen frequency, but not to the detriment of our technique. Indeed, optical rotatory dispersion measurements (the variation of optical rotation with wavelength) have proven extremely useful for isotropic samples and the same will surely be true for anisotropic samples.

Initial report from Referee 2 and our responses:

The paper by R. P. Cameron, U. Vogl and N. Trautmann proposes a new technique based on polarization interferometry to quantify the circular birefringence (optical rotation) in anisotropic samples. The work suggest that a new type of interference contrast microscopes (“chiral microscopes”) could use this technique for example to monitor the growth of chiral crystals.

The idea that the paper explores, an interference contrast that depends only on the optical activity and not on the linear birefringence, is very interesting and it is worth considering. The manuscript is very well written and the ideas are clearly presented.

We believe this to be a fair appraisal of our manuscript and thank Referee 2 for their kind words.

The only (major) problem that I find is that the real feasibility of the technique is not discussed. The entire paper is based on simulations and assuming ideal optical components and detectors, but being optical activity

on anisotropic materials “very hard to measure” it is desirable that any proposed technique does not disregard the experimental feasibility. I think that there are two main points that need discussion:

The contrast shown in Fig. 8 is really small. Note that all the grayscale levels are contained in just 0.01. Can imaging detectors detect this very small interferometric visibility in real experimental conditions of noise?

The basic setup in Fig. 1 contains four optical components (two mirrors and two beam splitters) which will introduce changes in the polarization of the beams that will be at least one order of magnitude larger than the optical rotation caused by a microscopic crystal. This is a particularly bad situation for beamsplitters where even does marketed “non-polarizing” have a rather large effect on polarization. Therefore it is expected that final interference contrast images can have very significant artifacts that, most likely, will be not treatable as systematic errors as they will also depend on the linear birefringence of the sample.

Recall that *two* possible versions of ICOA are described explicitly in our manuscript; ICOA-OR and ICOA-GOR. Let us consider each of these in turn below.

ICOA-OR

The basic setup shown in figure 1 was conceived of specifically for ICOA-OR and macroscopic samples like sample I, sample II and sample II (tilted), which show rather *large* interference contrasts, as is evident in figure 4. Slight component imperfections will not significantly affect the results in such cases.

While preparing our revised manuscript we noticed a sign error inherent to the basic setup described in section 1 of our original manuscript (an unwanted π phase shift). We have slightly modified the basic setup to rectify this error and adjusted section 1 of our manuscript accordingly.

ICOA-GOR

For ICOA-GOR and microscopic samples like sample III, sample IV and sample V, the basic setup depicted in figure 1 of our original manuscript is *not* suitable; to achieve a small shear distance ($\sim\mu\text{m}$, say) between sampling and reference fields, we envisage using a different setup.

We have significantly expanded section 2 of our manuscript by adding an explicit description of a basic setup suitable for ICOA-GOR and its use for microscopic samples. This setup does not have polarising beam splitters or mirrors like the setup described above for ICOA-OR and thus avoids the concerns of Referee 2. The small interference contrasts exhibited by sample III, sample IV and sample V should certainly be detectable using such a setup; for reference we note that an analogous lens-free microscope designed for standard DIC \cite{Terborg16a} has a sensitivity of *better* than $\sim 0.1\%$, which is to be compared with the $\sim 1\%$ sensitivity required by us for sample III, sample IV and sample V (this microscope is cited in the revised version of section 2).

In addition, we have added the following text to the caption of figure 9 of our revised manuscript to highlight the reduced scale used:

“For the sake of clarity we have plotted the intensity $\bar{I}^{\psi}_{\text{GOR}}$ for values in the range $[0.495, 0.505]$ rather than $[0, 1]$; black and white correspond to contrasts of $\bar{C}^{\psi}_{\text{GOR}} = 1\%$.”

We feel that further discussions of experimental feasibility are beyond the scope of this (already lengthy) manuscript; ICOA is a new technique and our primary aim here is simply to get the fundamental ideas across.

Minor issues

In Fig. 2 the line breaking between “100” and “ $2\pi c \dots$ ” can be confusing. It should be clearer in the legend and also in the caption that the green line has been multiplied by a factor 100.

As suggested, we have removed the line break and added the following sentence to the caption of figure 3 of our revised manuscript:

“For the sake of clarity the curves pertaining to γ have been amplified by a factor of 100 relative to those pertaining to α and Δ .”

For a better understanding of section 4a and Fig.2 reference should be made to the opposite signs of the optical activity tensors components of quartz, which produces different optical rotation signs depending on the angle under consideration and zero optical rotation at a specific angle.

We have added the following text to section 4a) of our revised manuscript:

“These dependences can be traced to the familiar optical rotation tensor g_{AB} ($g_{XX}=g_{YY}=\pm 5.82 \times 10^{-5}$ and $g_{ZZ}=\pm 1.296 \times 10^{-4}$), where the upper signs correspond to \mathcal{L} structure and the lower signs correspond to \mathcal{R} structure); we take

$$\gamma = \frac{\omega}{2c} \text{Big}[n^{(G)}_e - n^{(G)}_o \text{Big}] \sin \left(\tan^{-1} \left(\frac{G}{\text{Big}[n^{(0)}_e - n^{(0)}_o \text{Big}] \sqrt{n^{(0)}_e n^{(0)}_o}} \right) \right)$$

with

$$\begin{aligned} n^{(G)}_e &= \frac{1}{\sqrt{2}} \sqrt{n^{(0)2}_e + n^{(0)2}_o + \sqrt{\text{Big}[n^{(0)2}_e - n^{(0)2}_o \text{Big}]^2 + 4G^2}} \\ n^{(G)}_o &= \frac{1}{\sqrt{2}} \sqrt{n^{(0)2}_e + n^{(0)2}_o - \sqrt{\text{Big}[n^{(0)2}_e - n^{(0)2}_o \text{Big}]^2 + 4G^2}} \end{aligned}$$

using

$$\begin{aligned} G &= g_{XX} \sin^2 \vartheta + g_{ZZ} \cos^2 \vartheta \\ n^{(0)}_e &= \frac{n_e n_o}{\sqrt{n_e^2 \cos^2 \vartheta + n_o^2 \sin^2 \vartheta}} \\ n^{(0)}_o &= n_o, \end{aligned}$$

where $n^{(G)}_e$ is the extraordinary refractive index and $n^{(G)}_o$ is the ordinary refractive index.”

Response to Referees

for “Interference-contrast optical activity (ICOA): a new technique for probing the chirality of anisotropic samples and more” (RSOS-192201.R1)

Reproduced below is the decision letter from **Royal Society Open Science**, the report from **Referee 1** and the report from **Referee 2**, together with our responses to the referees.

Decision letter from **Royal Society Open Science**:

Dear Dr Cameron,

On behalf of the Editors, I am pleased to inform you that your Manuscript RSOS-192201.R1 entitled "Interference-contrast optical activity (ICOA): a new technique for probing the chirality of anisotropic samples and more" has been accepted for publication in Royal Society Open Science subject to minor revision in accordance with the referee suggestions. Please find the referees' comments at the end of this email.

The reviewers and Subject Editor have recommended publication, but also suggest some minor revisions to your manuscript. Therefore, I invite you to respond to the comments and revise your manuscript.

- Ethics statement

- Data accessibility

If you wish to submit your supporting data or code to Dryad (<https://eur02.safelinks.protection.outlook.com/?url=http%3A%2F%2Fdatadryad.org%2F&data=02%7C01%7CRobert.p.cameron%40strath.ac.uk%7C3582b2a4370040e0a0ff08d7dadacb72%7C631e0763153347eba5cd0457bee5944e%7C0%7C0%7C637218505562462212&data=pwYwjUuOhyOKmpnUzQXiL%2Bb20jrXQcvUt%2FUjqSaqMsQ%3D&reserved=0>), or modify your current submission to dryad, please use the following link:

<https://eur02.safelinks.protection.outlook.com/?url=http%3A%2F%2Fdatadryad.org%2Fsubmit%3FjournalID%3DRSOS%26manu%3DRSOS-192201.R1&data=02%7C01%7CRobert.p.cameron%40strath.ac.uk%7C3582b2a4370040e0a0ff08d7dadacb72%7C631e0763153347eba5cd0457bee5944e%7C0%7C0%7C637218505562472206&data=VGOLx%2BEUHPsFo0%2BS1zn%2F0YiJwpCNRXz%2BTquv8OXq%2FHM%3D&reserved=0>

- Competing interests

- Authors' contributions

- Acknowledgements

- Funding statement

Because the schedule for publication is very tight, it is a condition of publication that you submit the revised version of your manuscript before 16-Apr-2020. Please note that the revision deadline will expire at 00.00am on this date. If you do not think you will be able to meet this date please let me know immediately.

To revise your manuscript, log into <https://eur02.safelinks.protection.outlook.com/?url=https%3A%2F%2Fmc.manuscriptcentral.com%2Frsos&data=02%7C01%7CRobert.p.cameron%40strath.ac.uk%7C3582b2a4370040e0a0ff08d7dadacb72%7C631e0763153347eba5cd0457bee5944e%7C0%7C0%7C637218505562472206&sdata=hvMvjmXiXMQooA7K25LPBlyvEntPQUUkkmk53JZ0G9MM%3D&reserved=0> and enter your Author Centre, where you will find your manuscript title listed under "Manuscripts with Decisions". Under "Actions," click on "Create a Revision." You will be unable to make your revisions on the originally submitted version of the manuscript. Instead, revise your manuscript and upload a new version through your Author Centre.

Supplementary files will be published alongside the paper on the journal website and posted on the online figshare repository

(<https://eur02.safelinks.protection.outlook.com/?url=https%3A%2F%2Ffigshare.com%2F&data=02%7C01%7CRobert.p.cameron%40strath.ac.uk%7C3582b2a4370040e0a0ff08d7dadacb72%7C631e0763153347eba5cd0457bee5944e%7C0%7C0%7C637218505562472206&sdata=FTy8zH%2BP%2FBHkCb1LL3lWqHj8FC8KvUjc98W98ot799k%3D&reserved=0>). The heading and legend provided for each supplementary file during the submission process will be used to create the figshare page, so please ensure these are accurate and informative so that your files can be found in searches. Files on figshare will be made

available approximately one week before the accompanying article so that the supplementary material can be attributed a unique DOI.

Kind regards,
Lianne Parkhouse
Editorial Coordinator
Royal Society Open Science
openscience@royalsociety.org

on behalf of Dr Peter Munro (Associate Editor) and Miles Padgett (Subject Editor)
openscience@royalsociety.org

Associate Editor Comments to Author (Dr Peter Munro):

Please consider making the minor revisions suggested by the second reviewer.

Report from Referee 1 and our responses:

The manuscript is accepted in present form.

We thank Referee 1 for their recommendation that our manuscript be accepted for publication.

Report from Referee 2 and our responses:

Authors have done a good revision of the manuscript and they have answered clearly to the questions raised by the reviewers. I specially appreciate the more clear explanation of the ICOA-GOR method. I think the paper can be accepted for publication.

We thank Referee 2 for their recommendation that our manuscript be accepted for publication.

A couple of minor additional questions I have (authors may optionally want to consider them):

- The lens free method proposed for ICOA-GOR seems to have not very large lateral resolution and this can be contradictory with the examples of microscopic quartz crystals provided.

Referee 2 is correct in that the lens-free microscope described in \cite{Terborg16a}, for example, has a low lateral resolution of 35 μ m, limited by diffraction and other such effects. These effects are *not* included in our manuscript, as they are inherently neglected by the Jones vector formalism we have used. We have acknowledged this omission in section 5 and pledged to describe such effects as well as more realistic microscope designs in *future* works. We note here, however, that it *is* possible to obtain the desired shear and resolution for ICOA-GOR simultaneously for most samples of interest; the lateral resolution of 35 μ m quoted in \cite{Terborg16a}, for example, can be improved by decreasing the distance between sample and detector.

Note that our manuscript is completely self-consistent at the level of description we have employed.

- In the context of ICOA-GOR authors should probably cite other papers in which interferometric single-beam setups have been used to measure optical activity (e.g. [10.1002/mop.4650060706](https://doi.org/10.1002/mop.4650060706) and [10.1364/OE.27.006746](https://doi.org/10.1364/OE.27.006746))

We thank Referee 2 for bringing these references to our attention. In response, we have modified the following text to section 1

“ICOA-OR and ICOA-GOR are distinct from all polarimetric techniques known to the authors at the time of writing, including ... existing polarisation interferometry techniques \cite{Echarri93a} ...”

and the following text to section 3

“Interestingly, the initial optical field here (which sports one-dimensional helicity fringes) is essentially the same as the optical field proposed by one of the authors for exerting discriminatory optical forces on chiral molecules \cite{Cameron14a}; a phenomenon recently demonstrated in the laboratory for chiral liquid crystal microspheres \cite{Kravets19a}. It is also essentially the same as the optical field employed in the new technique of snapshot circular dichroism, which might enable faster measurements of circular dichroism than is possible using traditional circular dichroism spectrometers \cite{Arteaga19a}.”